# Investigating the Diversity of Tuberculosis Spoligotypes with Dimensionality Reduction and Graph Theory

**DOI:** 10.3390/genes13122328

**Published:** 2022-12-10

**Authors:** Gaetan Senelle, Christophe Guyeux, Guislaine Refrégier, Christophe Sola

**Affiliations:** 1FEMTO-ST Institute, UMR 6174 CNRS, Université de Bourgogne-Franche-Comté, Burgundy-Franche-Comte, 90000 Belfort, France; 2Université Paris-Saclay, CNRS, AgroParisTech, Ecologie Systématique Evolution, 91405 Gif-sur-Yvette, France; 3Université Paris-Saclay, 91190 Saint Aubin, France; 4IAME Laboratory, UMR 1137 INSERM, Université Paris-Cité, 75870 Paris, France

**Keywords:** *Mycobacterium tuberculosis*, CRISPR, dimensionality reduction, graph theory

## Abstract

The spoligotype is a graphical description of the CRISPR locus present in *Mycobacterium tuberculosis*, which has the particularity of having only 68 possible spacers. This spoligotype, which can be easily obtained either in vitro or in silico, allows to have a summary information of lineage or even antibiotic resistance (when known to be associated to a particular cluster) at a lower cost. The objective of this article is to show that this representation is richer than it seems, and that it is under-exploited until now. We first recall an original way to represent these spoligotypes as points in the plane, allowing to highlight possible sub-lineages, particularities in the animal strains, etc. This graphical representation shows clusters and a skeleton in the form of a graph, which led us to see these spoligotypes as vertices of an unconnected directed graph. In this paper, we therefore propose to exploit in detail the description of the variety of spoligotypes using a graph, and we show to what extent such a description can be informative.

## 1. Introduction

The so-called Clustered Regularly Interspaced Short Palindromic Repeats (CRISPR) of the *Mycobacterium tuberculosis* complex (MTC) is constituted by a sequence of 36 nucleotides called direct repeat (DR) interspersed with single “spacer” sequences ranging from 25 to 45 nt [1]. This locus, identified in 1993 [2] in two MTC genomes (namely, *M. bovis BCG* and *M. tuberculosis* H37Rv), yielded 43 spacers. Analysis of presence of each of these spacers was called “spoligotyping” [3], and this popular technique has help to the understanding of the MTC diversity [4,5]. The more recent appearance of whole genome sequencing has revealed the existence of six major lineages comprising numerous sub-lineages [6,7,8,9,10], which have frequently been found to be in phase with previously discovered spoligotype patterns, despite the presence of convergent or imprecise events that occasionally confound the information carried by the latter.

The CRISPR-Cas locus of type III-A group [11] keeps a immune activity but does not adapt anymore and can actually be partly or totally removed in rare sublineages [12,13]. The study of the variability of this region has started in 2000 [14], emphasizing IS*6110* insertions and spacer variations depending on the lineages. However the good understanding of the evolutionary dynamics of this region still needs further investigations on a big dataset. Basic spoligotyping applied in vitro only lists the presence of 43 spacers [3], which ignores the richness of this locus: a change on the spacer order, the existence of duplication, the insertions of IS*6110* and their locations, or even mutations in spacers or DR. First generation in silico methods (SpolPred [15], SpoTyping [16]) that have been designed to obtain spoligotypes from genome sequences [15] share the same limitations as in vitro approaches. This is the origin of the emergence of new technologies like CRISPRbuilder-TB [17], whose aim is to capture the whole diversity of this locus.

This paper recalls our Spolmap proposal, presented at the 9th International Work-Conference on Bioinformatics and Biomedical Engineering (IWBBIO 2022 [18]). The idea is to see each spoligotype as a list of holes, to which we can univocally associate a vector in a high dimensional space, and then to use t-SNE [19], a powerful dimensionality reduction technique, to project this point cloud in a plane. The visualization of the spoligotypes in this plane reveals coherent patterns in terms of lineages and sub-lineages, and various intriguing clusters just waiting to be studied further. Above all, a graph structure appears, which encourages us to see the spoligotypes as vertices of an ad hoc graph. The objective of this extended version of the IWBBIO 2022 article is to describe how this graph is formed, to propose some study paths using tools from graph theory (depth, order, connectedness*…*), and to extract in return some elements for reflection concerning the evolution of the bacteria, the representativeness of spoligotypes, etc.

The rest of this paper is organized as follows. Section 2 is devoted to the first proposal, namely the Spolmap, an enriched visualization of CRISPR diversity approach presented at IWBBIO 2022. This representation leads to the proposal of the SpolGraph made in Section 3, which is fully defined, studied, and for which the consequences on the evolution of *M. tuberculosis* are deduced.

## 2. Spolmap: Enriching the Visualization of CRISPR Diversity

As a reminder, Spolmap was built on 68-spacers spoligotype patterns derived from WGS public archives (Short Read Archives) by simplifying the outputs from full reconstruciton of CRISPR locus (CRISPRbuilder-TB). This simplification ignores SNPs in DR and spacers and includes spacers when present even if an insertion sequence would impede its detection in in vitro spoligotyping. To proceed to spoligotypes’ simplification we took into account the fact that they evolve by deletion of spacers, i.e., we took into account gaps. The following paragraph explains the main steps of our procedure. More details can be found in the methods and by analyzing the script made available at https://github.com/cguyeux/spolmap.git (accessed on 7 December 2022).

Suppose that the investigated spoligotypes contain *N* gaps, that is (15, 26); (30, 34); (51, 60) for the first pattern of Figure 1. The first stage is to map each pattern in a space of *N*-dimension. To do so, gaps are sorted lexicographically, and an integer is associated to each gap, depending to such an order. We then associate a vector to this spoligotype as follows. If there is a gap in the considered position, we put an 1, else we put 0. An N-size binary vector is thus obtained, and each different pattern has a specific position in that space. By doing so, closed points have similar spoligotypes. The last step consists of a dimentionality reduction thanks to the t-SNE approach [19].

An example of what Spolmap can lead to is shown in Figure 2 in the MTC case. As can be seen, there are as many clusters as there are lineages, with sub-clusters associated with sub-lineages. Some lineages are very well separated and present a really pure cluster, such as lineages 5, 6 and 7. We also find, in the upper right part, lineages 2 to 4, and in the lower left part, lineages 1, 5, 6 and animal, and we know that these two subgroups are phylogenetically separated. The clusters of lineages 1 to 4 extend to the center of the cloud, arguing for a common origin of the tuberculosis complex, whose common ancestor could be *M. canettii* [20].

The SNP-based lineage data and the spoligotype hole data are strongly correlated, arguing for a co-occurrence of these two evolutionary mechanisms. In some sublineages, however, the corresponding subcluster is only partially colored, suggesting a poor definition of said sublineage (an overly restrictive lineage SNP). We also see a whole big gray cluster with a few green dots in it, which would tend to show our very poor knowledge of animal TB.

## 3. Spolgraph: Adding a Graph Structure to the Spolmap Representation

### 3.1. Biological Reasons to Be Interested in a Graph Structure

The representation in Figure 2 shows a cluster structure, in which the spoligotypes seem to explode, deriving from ancestors located in the clusters. This is in line with the mode of evolution of the CRISPR locus in *M. tuberculosis*, as we have specified in [1,17].

It is known that in a number of bacteria, this locus is made up of pieces of bacteriophage viruses that have been “memorized” by the genome following a non-lethal aggression, each new aggression leading to the addition of a spacer (phage sequence) in the CRISPR. However, *M. tuberculosis* has the particularity of “hiding” in human cells, and thus it is no longer subjected to aggression from viruses, the cell protecting it. This would explain why the CRISPR locus does not seem to be active in *M. tuberculosis*. Since it is no longer under attack, the bacterium does not store new pieces of phage in the form of spacers, which explains why the list of these spacers is finite and small. There are indeed only 68 spacers currently known in *M. tuberculosis*, and our work published in [1] tends to show that no new spacer is to be discovered. These 68 spacers would be the content of the CRISPR that the ancestor of *M. tuberculosis* would have, when it went to hide inside the cell.

It is also known that the *M. tuberculosis* genome contains a number of copies of the IS*6110* insertion sequence, and that the CRISPR of the vast majority of strains has an insertion in its middle. We have also shown in [1] that the insertion of a second IS*6110* in this locus frequently leads to recombination between these two IS, deleting all the spacers located between these two insertion sequences. We also mentioned the possibility of recombination between two direct repeats, again leading to the deletion of all spacers between these two repeats.

Such a description is based on certain hypotheses which, if they seem reasonable and probable, are nonetheless unproven for the moment. They may never be proven. For example, it is difficult to prove that the 68 spacers are bits of mycobacteriophages virus, and even though there are nice nucleotide similarities, it is difficult to show that these similarities do not occur by chance, because the spacers are so small. But if these reasonable assumptions are true, we would have a locus that evolves mainly by loss of spacer blocks, due to recombination between repeated sequences. These losses of bits of genomes would not be deleterious, since the locus would no longer be operational, and they would therefore be conserved during the evolution of these bacteria.

Such a mechanism described is in line with observations. They explain well the shapes of the spoligotypes, rich in successive spacer holes, the number and size of the holes increasing from one lineage to its sub-lineages. This explains the relevance of the study of spoligotypes as a lineage intelligence tool, and encourages a more in-depth study of the links between motifs, as we will do below.

### 3.2. Introducing the Graph Structure

As reminded above, the CRISPR of *M. tuberculosis* evolves by loss of individual or successive blocks of spacers. Again, we established a link between two spoligotypes, when one can go from the first to the second by the deletion (editing type operation) of one and only one successive spacer block. The vertices of the directed graph would be the spoligotypes, an arc connecting a father vertex to a son when we can go from one to the other by exactly one deletion mechanism or deletion step, see Figure 3. Clearly, at the root of the tree, we would find the full pattern corresponding to the presence of the 68 spacers.

At this level, we must make an important remark. The original spoligotype does not consist of the presence/absence of the 68 known spacers in *M. tuberculosis*, but of only 43 “historical” spacers. These historical spacers, mainly based on the content of the H37Rv reference (lineage 4.9) are, strictly speaking, what is usually represented as a spoligotype, and what is obtained in vitro by the usual molecular biology typing techniques. Studying the 68-spacer version would clearly make more sense, biologically speaking, and one is closer to the mechanisms of genomic recombination when working on size 68 spoligotypes. However, this is not the choice that will be made here. We need data, spoligotypes with associated lineages, to build our graph and study it. In the literature, the complete 68-element spoligotype is almost never studied, and the existing historical databases like SITVIT2 [21] are all based on the 43-spacer version. Also, existing tools such as SpoTyping or SpolPred only produce spoligotypes of size 43. To conclude, although the 68-element version is richer and reflects reality, we will not be able to study it here, in the absence of existing databases. The development of a python script using also Blast to check the presence/absence of not only the 43 spacer list but the 68 one - and even the 94 list including the 26 from *canetti* would help.

From the adjacency relation described above, we filled our directed graph, called SpolGraph, from the data available on the SITVIT2 website, and from additional data kindly provided by Jody Phelan (LSHTM, London School of Hygien and Tropical Medicine, personal communication). We discarded spoligotypes for which the lineage was not known, because we want to extract knowledge from the SpolGraph. Also, when several lineages were associated with the same spoligotype, we kept only the first lineage recorded in the list. The result is a graph with 8455 vertices (the spoligotypes), labelled by the lineage, and 8694 edges. The root is the spoligotype consisting only of ones, mostly associated to the Beijing 2.1 or to the L1 lineage.

The implementation was done in Python 3 [22], with the networkx library [23] for the management of directed graphs. The spoligotypes are described and stored in the form of hole lists. For example,


  1110111111111111111100001111111100001111111


becomes:


  [(4, 4), (21, 24), (33, 36)]


This transcription can be obtained as follows:



The fact of knowing if a spoligotype is the father of another is simply written in Python:



### 3.3. Investigating the SpolGraph

An analysis of the variety of the vertex labels yields the results of Table 1. The lineages are clearly not represented in the same way: lineage 4 exceeds 5000 nodes (64.44%), while lineage 7 has less than a dozen nodes. This unfortunate fact is however representative of the current study of tuberculosis lineages: the one found in Europe and in the USA (lineage 4) is much more studied than those found in Africa (lineages 5 to 7), and this is reflected in the sequencing and in the databases.

#### 3.3.1. The Lineages in the Graph

When you look at the diversity of peaks as a whole, the spoligotypes are generally quite full of holes. More exactly, the average is 4.07 holes, for a rather low standard deviation of 1.62. However, this average hides some disparity by lineage, cf. Table 2. These results are interesting, insofar as these holes can be interpreted biologically: a higher number means either a longer history (the time for these recombinations to take place), or a more eventful history (a higher mutation rate). For example, the African lineages L5 and L6 are known to be currently “calmer” than the other lineages, evolving at a slower pace, and we see that they have on average fewer holes than the other lineages. This is even more obvious for lineage 7, which is the lineage with fewer holes on average. If we assume that this average is significant in spite of the small number of strains, then it is related to the fact that this lineage is known to evolve little, and is only found in Ethiopia. One can also be surprised by the relatively high average for lineage 2, knowing that the modern Beijing (2.2) have all lost the vast majority of their CRISPR. This average of 3.06 holes can however be explained by a high activity of IS*6110* in this lineage.

#### 3.3.2. The Connectivity of the Graph

We can then ask ourselves a first question, coming from the theory of graphs and which has a biological interest, namely is the graph thus constructed connected? If so, by definition, for any pair of vertices, there is a path in the graph leading from the first to the second (the graph is “in one piece”). Such a connectedness would mean that we have collected all naturally occurring spoligotypes, and that we can insert each spoligotype in its evolutionary history from the initial pattern: each deletion step would be documented by an existing strain. Conversely, a graph with many unrelated components means that many spoligotypes are missing, either because not enough strains have been sequenced (poor spatial representativeness), or because intermediate spoligotypes have disappeared over time (poor temporal representativeness).

The connected_components command applied to the undirected graph leads to an impressive number of 2779 connected components. While this may seem disappointing at first, a closer look at the components shows that this is not the case. First of all, the largest component has 5028 nodes, i.e., about 60% of the spoligotypes. It also contains the ancestral pattern. For these 5028 patterns, we are therefore able to detail, step by step, all the recombinations that may have taken place historically. Having a first component of such a size also allows us to think that we finally have a fairly good representation of the existing spoligotypes, and that these recombinations are not very old (they did not have time to disappear).

At the other end of the spectrum, 2635 of the 2779 related components consist of a single vertex. For each of these spoligotypes, at least two recombinations from another known spoligotype (no father) were required, and they did not give rise to any other known spoligotype (no son), which is unlikely. Also, some of these patterns must be erroneous, and the constitution of the SpolGraph is also an opportunity to detect “suspicious” patterns.

The intermediate components have sizes ranging from 2 to 244 nodes. Their size decreases rapidly, and only 10 components have more than 10 spoligotypes, but these 11 largest components represent only 2/3 of the patterns. An interesting point about the small components is that they are generally homogeneous in lineages. For example, the component consisting of the spoligotypes:


0101101000001110111111111111111111111100000



1101101000001110111111111000000111111100000



1101101000001110111111111111110000001100000



1101101000001110111111111111111011111100000



1101101000001110111111111111111111111100000



1101111000001010111111111111111111111100000



1101111000001110111111111111111111111100000


contains only *M.bovis*. Moreover, the small components are over represented in lineages 3, 5, 6, 7 and animal, that is to say in the least studied lineages, which explains this. These components not reduced to a point should be able to be linked to the larger component, through one or more missing motifs (either because they have not been found yet, or because they no longer exist).

#### 3.3.3. In-Depth Study of Certain Aspects of the Graph

Other interesting elements can be easily extracted from this description of the spoligotypes in graph form.

First, the maximum depth of the graph is 8, corresponding to a maximum of 8 theoretical recombination events from the initial pattern. An example of such a path in the graph is given below:


1111111111111111111111111111111111111111111



1111111111111111111111111111111100001111111



1111111111111111111111111111111100001110111



1111111111111111111111111111111100001110110



1100111111111111111111111111111100001110110



1100111111111111001111111111111100001110110



1100111011111111001111111111111100001110110



1100111010111111001111111111111100001110110



1100111010111111001011111111111100001110110


Somewhat strangely, the patterns 2 and 3 visited are from lineages 4.1.2.1 and 4.2.2 respectively. The other spoligotypes are all from lineage 4.6.1.1, which shows a real coherence and a strong link between lineage according to single nucleotide polymorphism (SNP) and spacer holes. Evolutionary scenarios can also be established. Thus, above, the last 6 recombinations occurred after the mutation (SNP) chosen to define lineage 4.6.1.1, when the first two motifs appeared early in the history of lineage 4, and remained preserved until lineages 4.1.2.1 and 4.2.2.

The graph has a total of 34 paths of maximum length 8. This high depth of 8, with its frequent appearance (34 paths) indicates a high recombination activity at the CRISPR level. But it is not evenly distributed across all lineages. Indeed, only 4.6.1.1 and 1.1.2 are found at the leaf level of these 34 paths, with a consistent history in terms of sublineages from the original motif. However, it remains to be understood what is special about these two sublineages.

Another remarkable point is that the degrees of the vertices (the number of child nodes) vary greatly. The original pattern has 140 child nodes, but it does not have the highest degree. The latter is the spoligotype:


1111111111111111111111111111111100001111111


which has the double (280) of son nodes. The third highest order vertex is:


1111111111111111111111111111111100001110111


He is a convergent signature of both lineages 4.2.2 and 4.6, and has 123 sons. One reason for such a high degree may be the over-representation of a given lineage. Indeed, these lineages are very well studied, and there are 805 different spoligotypes of these lineages in SITVIT2. This argument does not hold, however, when we recall that the previous study shows that we have a fairly good knowledge of the diversity of spoligotypes. In other words, it is not because T3-ETH-Osa-Tur (L4.2.2) is studied a lot that we have a lot of different spoligotypes, but because recombinations at the CRISPR level are very frequent in this sublineage. Another reason could be that some loss of spacers appeared independently many times, for example the absence of spacer 40. A manual reconstruction of the CRISPR-CAS locus of various strains of this lineage shows, moreover, a large number of sites of insertion of IS*6110* in this locus, leading to such a recombination dynamics. Moreover, there are “only” 62 different spoligotypes of 4.2.2, which shows that the link between high degree of a node and high number of motifs in the associated lineage is not direct.

## 4. Conclusions

In this paper, we have extended the work presented at IWBBIO 2022 on the 2D representation called SpolMap to a graph structure from which it is inspired. SpolMap, based on t-SNE, indeed allowed to highlight genome clusters consistent with *M. tuberculosis* lineages, the whole linked by a graph skeleton. This structure, called SpolGraph, is compatible with genome evolution, insofar as a node (a spoligotype) is the father of another one if, and only if, we pass from one to the other by a recombination (between IS*6110* or DR) at the CRISPR level. This different way of looking at the diversity of spoligotypic patterns allowed us to deduce various information about the representativeness of the current databases, about the rate of recombination per lineage, etc. This code is available at https://github.com/cguyeux/spolmap.git (accessed on 7 December 2022).

In future work related to SpolMap, we wish to propose versions for *M. tuberculosis*, *Salmonella* and *Legionella*. A discussion could also be built speaking about relation between structure and function, especially when it comes to transcription and small RNAs. We then wish to integrate all available genomes (more than 100,000 genomes in the case of *M. tuberculosis*), and then to search for unknown lineages. We intend to integrate this representation in a larger and complete tool, including for example the determination of lineages and MIRU-VNTRs in *M. tuberculosis*. Finally, the ways to extend SpolGraph to 68 spacer based spoligotypes will be investigated, and the resulting graph will be more systematically studied. Such developments may be facilitated in the future by the advent of long read sequencing techniques, providing complete CRISPR structure.

## Figures and Tables

**Figure 1 genes-13-02328-f001:**
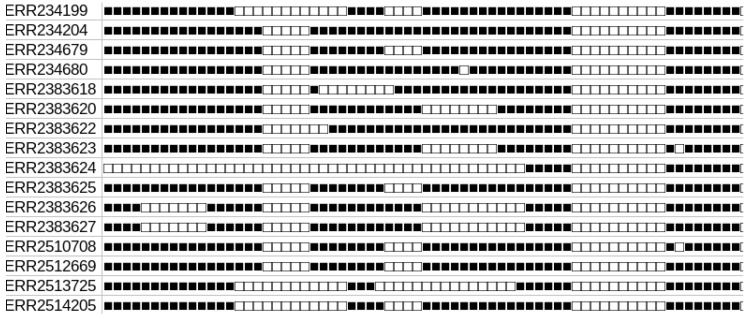
Example of spoligotypes of Lineage 5, defined by their accession numbers.

**Figure 2 genes-13-02328-f002:**
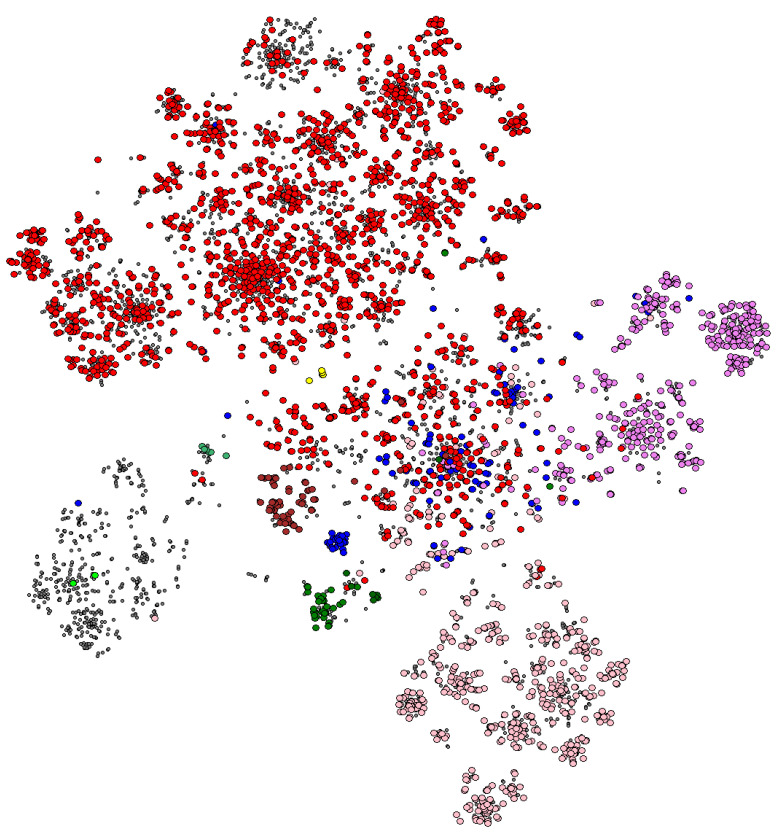
A 2D visualization of the MTC’s spoligotypes. Each point corresponds to one strain (in WGS genome form), while the color of these points is made according to the strain lineage as defined by the presence of classification SNPs.

**Figure 3 genes-13-02328-f003:**
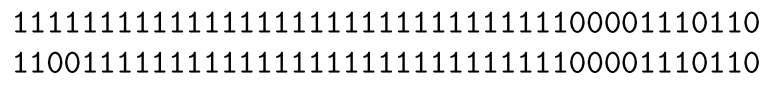
Two adjacent nodes, the father at the top, the child at the bottom (ancestor and offspring).

**Table 1 genes-13-02328-t001:** Nodes per lineage in the SpolGraph.

Lineage	Number of Nodes
1	1415
2	314
3	970
4	5449
5	98
6	120
7	9

**Table 2 genes-13-02328-t002:** Mean and standard deviation of holes in spoligotypes per lineage.

Lineage	Mean	Standard Deviation
1	4.60	1.50
2	3.06	1.77
3	3.68	1.35
4	4.07	1.65
5	3.93	1.30
6	3.47	1.08
7	2.89	0.78

## Data Availability

SITVIT2 at http://www.pasteur-guadeloupe.fr:8081/SITVIT2/ (accessed on 7 December 2022).

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
