# Peer review of "Investigating the Diversity of Tuberculosis Spoligotypes with Dimensionality Reduction and Graph Theory"

_genes, 2022, doi:10.3390/genes13122328_

Round 1

Reviewer 1 Report

The paper presents a new algorithm SpolMap to improve and visualize spoligotyping of M. tuberculosis isolates. The authors said in the conclusion: “In future work related to SpolMap, we wish to make this tool accessible through a neat interface, and propose versions for tuberculosis, salmonella and legionella.” It is not acceptable. First, publish at least a beta-version of the program, for example at GitHub (https://github.com/), then publish your paper without making any promises. I do not think that spoligotyping is applicable for all bacteria as many of them do not possess either CRISPR inserts or any other tandem repeats.

Spoligotyping was a useful tool in Mtb studies a few decades ago as it allowed amplification of the tandem repeat regions for further analysis either by gel electrophoresis or by Sanger sequencing. With a wide use of NGS, SNP detection became a more preferable approach of Mtb lineage identification for many obvious reasons. Consider the Web-application PhyTB (http://pathogenseq.lshtm.ac.uk/phytblive/index.php), which allows an unambiguous Mtb lineage identification using raw FASTQ files without a need for blasting or genome assembly.

The authors made an attempt to improve prediction of spoligotypes by their program, but ended up with a recommendation to finalize predictions manually. In other words, they failed. Fig. 2 is not impressive also as many lineages overlap partly or completely.

3rd Generation long-read sequencing technologies (SMRT PacBio and Oxford Nanopore) may bring a new life to spoligotyping as they became affordable and as accurate as the Illumina. These new technologies may require new tools, but the authors ignored these new developments.

I did not understand the authors’ hypothesis of the biological mechanisms of evolution of spoligotypes. As long as the CRISPR insert remains inactive in the modern Mtb isolates, they may only lose their repeats. The authors said something about a possibility to gain repeats by recombination. What kind of recombination is meant? Did the authors suggest a possibility for recombination between different lineages or sublineages of Mtb? If the authors can prove this hypothesis, it will be interesting to publish. At the moment it sounds like a vague hypothesis.

Neither the application of graphs, nor the provided Python, nor the shown outputs are novel or of scientific merit.

Comments regarding the style of the paper:

1.       The current introduction can be mostly removed and the section 2 “Basic recall” can be used as an introduction.

2.       Section 4 with the technical description of the SpolMap algorithm should be removed from this paper and either directed to a journal dealing with bioinformatics algorithms (my personal opinion – it won’t be accepted) or published online on the SpolMap Web-site at GitHub (it would be the best). Generally, I do not think that BLASTN is the best option for this algorithm as it ignores base call quality values presented by FASTQ files.

3.       Page 3 “Firstly, we have to download the genomes of interest, in the form of a Sequence Read Archive (SRA), …” – why not to say that the input files must be in FASTQ format. There should be no difference whether these files are from SRA or generated de novo by sequencing. Respectively, in the next abstract “SRA file” – FASTQ file?

4.       Bullet-pointed descriptions of Fig. 2 on pages 5-6 should be moved to the figure legend.

5.       Page 7, “Many conclusions can be drawn from this point cloud obtained from the spoligotypes.” – remove this sentence, go straight to the conclusions.

6.       Page 14, “a high concentration of IS6110 in this locus” – definitely not a “concentration”. May be a big number of IS6110, but before it was said that 2 IS6110 was the maximum. Is it a big number?

Author Response

Please, see the attached pdf.

Reviewer 2 Report

I would like to congratulate the authors of this article for their clarity in presenting the results and conclusions. Perhaps I have not understood it well, but I have not been able to know how many strains of the different lineages have been studied and what their origin has been. Other than that, I find it to be a very interesting article.

Author Response

Thank you for this very positive remark.

(We have used all the spoligotypes referenced at SITVITWEB2)

Reviewer 3 Report

The authors report a graph based and dimensionality reduction method to detect M. tuberculosis based on spacer signatures. While the work is a good concept, it is presented as a straightforward talk instead of a research paper.

The authors need to structure it properly, edit the language scientifically at places, as well as language.

Several places need references and they should format it according to journal so that I can make more sense of it.

The authors say that they intend to develop an interface to this method in future. If we can have this now, I could have run and checked more details. The paper would have become stronger.

Author Response

The authors report a graph based and dimensionality reduction method to detect M. tuberculosis based on spacer signatures. While the work is a good concept, it is presented as a straightforward talk instead of a research paper.

The authors need to structure it properly, edit the language scientifically at places, as well as language.

Answer: The structure has been updated and an English speaker has reviewed the whole article.

Several places need references and they should format it according to journal so that I can make more sense of it.

Answer: references have been updated, and we now use the journal format.

The authors say that they intend to develop an interface to this method in future. If we can have this now, I could have run and checked more details. The paper would have become stronger.

Answer: The software is now accessible online via github.

Thank you for all these interesting remarks.

The authors

Round 2

Reviewer 1 Report

The paper was significantly improved and the program code was made publicly available. All my previous questions were addressed. I support publishing this paper.